# Functional Study and Efficient Catalytic Element Mining of CYP76AHs in *Salvia* Plants

**DOI:** 10.3390/molecules28124711

**Published:** 2023-06-12

**Authors:** Zhenyu Zhao, Dongfeng Yang, Juan Guo, Xiuyu Liu, Qishuang Li, Ping Su, Jian Wang, Ying Ma, Luqi Huang

**Affiliations:** 1State Key Laboratory of Dao-di Herbs, National Resource Center for Chinese Materia Medica, China Academy of Chinese Medical Sciences, Beijing 100700, China; zhaozhenyu9607@163.com (Z.Z.); guojuanzy@163.com (J.G.); suping120@163.com (P.S.); wangjian2021@126.com (J.W.); 2Key Laboratory of Plant Secondary Metabolism and Regulation in Zhejiang Province, College of Life Sciences and Medicine, Zhejiang Sci-Tech University, Hangzhou 310018, China; ydf807@sina.com; 3College of Pharmacy, Henan University of Chinese Medicine, Zhengzhou 450008, China; liuxiuyuzy@163.com; 4School of Traditional Chinese Pharmacy, China Pharmaceutical University, Nanjing 210009, China; liqs556@163.com

**Keywords:** *Salvia*, CYP76AH, P450, biosynthesis, functional gene

## Abstract

*Salvia* is a large genus with hundreds of species used in traditional Chinese medicine. Tanshinones are a highly representative class of exclusive compounds found in the *Salvia* genus that exhibit significant biological activity. Tanshinone components have been identified in 16 *Salvia* species. The CYP76AH subfamily (P450) is crucial for the synthesis of tanshinone due to its catalytic generation of polyhydroxy structures. In this study, a total of 420 CYP76AH genes were obtained, and phylogenetic analysis showed their clear clustering relationships. Fifteen CYP76AH genes from 10 *Salvia* species were cloned and studied from the perspectives of evolution and catalytic efficiency. Three CYP76AHs with significantly improved catalytic efficiency compared to SmCYP76AH3 were identified, providing efficient catalytic elements for the synthetic biological production of tanshinones. A structure–function relationship study revealed several conserved residues that might be related to the function of CYP76AHs and provided a new mutation direction for the study of the directed evolution of plant P450.

## 1. Introduction

*Salvia* is a vast genus of medicinal plants distributed worldwide, comprising at least 1000 species, 78 of which are native to China. However, variations have been observed in the quantity and type of secondary metabolites found in different *Salvia* species [1]. The most characteristic compounds in *Salvia* are abietane diterpenoids, which comprise tanshinones and have a variety of biological activities, particularly antioxidant benefits and pharmacological actions for treating heart disease [2]. *Salvia* species are used in a variety of Chinese herbal medicines, including Danshen (*Salvia miltiorrhiza*) and Diandanshen (*Salvia yunnanensis*); most have been reported to accumulate tanshinones. Tanshinones are one of the main active ingredients of Danshen in the treatment of cardiovascular diseases, with various pharmacological activities [3]. Tanshinone IIA is a well-known monomer that exhibits substantial biological activity in the treatment of neurological illnesses and significant pharmacological effects in the treatment of cardiovascular diseases [4]. The differences in metabolic components are significantly correlated with the types and expression levels of biosynthetic pathway genes. Therefore, *Salvia* plants are very suitable for conducting correlation studies on chemical diversity and gene differences due to their large population and wide distribution.

CYP450 (cytochrome P450) is essential in the biosynthesis of natural active ingredients in plants and participates in the biosynthesis of most terpenoid active ingredients [5]. Due to the high degree of oxidation of tanshinones, multiple P450s are required to participate in this biosynthesis pathway, and CYP76AH subfamily P450s play a significant role in the synthesis of tanshinones [6]. The biological functions of CYP76AH subfamily genes in Labiatae medicinal plants have been continuously reported in recent years. SmCYP76AH1 in *S. miltiorrhiza* has been identified as a ferruginol synthase that catalyzes the hydroxylation of C12 of miltiradiene [7]. The biological functions of SmCYP76AH3 and SmCYP76AK1 have been screened with engineering yeast construction and mature hairy root genetic transformation systems [8]. SmCYP76AH3 is a hybrid catalytic P450 that can catalyze the hydroxylation of ferruginol at C11 and the carbonylation of ferruginol at C7 to generate 11-hydroxy ferruginol, sugiol, and 11-hydroxy sugiol. The carbonylation function of C7 leads to the possible extension of the biosynthetic pathway of tanshinone to another branch, thus forming a complex catalytic network (Figure 1). SmCYP76AK1 is one of the key enzymes in the biosynthesis pathway of tanshinone compounds, with a specific C20 hydroxylation function [8]. CYP76AH4, CYP76AH22-24, and CYP76AH57 have been identified in *S. fruticosa* and *Rosmarinus officinalis*, and their catalytic functions are similar to that of SmCYP76AHs [9,10]. In addition, *CfCYP76AH15*, *CfCYP76AH11*, and *CfCYP76AH16*, which were cloned from the cortex of *Coleus forskohlii* and transiently expressed in tobacco, can catalyze the multistage reaction of the precursor 13R-mannoyl oxide to produce various labdane diterpenoids [11].

Miltiradiene is the biosynthetic precursor of many active diterpenes [12,13,14]; therefore, it is very important to study the catalytic function of the P450s that start the first modification steps of miltiradiene. Miltiradiene undergoes hydroxylation at the carbon atoms C7, C11, and C12 to form a number of different chemicals with the catalytic action of CYP76AHs [8,9,10]. In this study, CYP76AH homologous genes were isolated and subjected to tertiary structure comparison, bioinformatics study, and evolutionary analysis. Fifteen CYP76AH genes from ten different species of *Salvia* were cloned, and miltiradiene was used as the substrate for the enzymatic reaction. The catalytic efficiency of producing different C11, C7, and C12 oxidation products was analyzed. All of the enzyme genes had distinct effects and produced different products, and key enzyme genes with higher catalytic efficiency than SmCYP76AH1 and SmCYP76AH3 were discovered.

## 2. Results

### 2.1. Chemical Constituents of Salvia

The biosynthetic pathway of tanshinones was characterized in Danshen. Several components, including ferruginol, sugiol, miltirone, cryptotanshinone, 2-isopropyl-8-methylphenanthrene-3,4-dione (Ro-09-0680, abbreviated as R09 in this article), and dihydrotanshinone I, were shown to be involved in tanshinone biosynthesis (Figure 1). According to previous research, the content of intermediates and end products in the tanshinone metabolism pathway varies significantly between species (Figure 2). Sixteen distinct species from the *Salvia* genus were selected for further analysis, representing high, medium, and low levels of tanshinone based on the findings of previous studies. Among the *Salvia* species studied, *S. honania* had the greatest ferruginol concentration at 0.05352%, followed by *S. digitaloides* at 0.02534%, both of which were much greater than the ferruginol content in Danshen. *Salvia aerea* had the highest content of sugiol at 0.0186%, followed by *S*. *digitaloides* at 0.01362%. The proportion of miltirone content in *Salvia* plants is significantly higher than that in ferruginol and sugiol and shows a high level overall. The tanshinone content of the six species was higher than that of *S. aerea*, *S. castanea*, *S. digitaloides*, and *S. miltiorrhiza*. The cryptotanshinone content in *S. milliorrhiza* was relatively high. The R09 content of various species in the *Salvia* genus was generally lower than that of miltirone and cryptotanshinone, and *S. digitaloides* had the highest R09 content. The dihydrotanshinone I content in *S. digitaloides* was the highest. The results of this study suggest that there may be more alternative varieties of *S. miltiorrhiza* in *Salvia*. At the same time, it provides a material basis for the excavation of diterpenoid biosynthetic elements and the study of catalytic mechanisms in *Salvia*.

### 2.2. Phylogenetic Analysis of Candidate CYP76AH Subfamily Genes

CYP76Ahs have been reported as the first CYP450s involved in the biosynthesis of abietane-type diterpenoids, which catalyze the carbon skeleton miltiradiene to produce ferruginol or further oxidation to sugiol [7]. The evolution of CYP76AH plays an essential role in diterpenoid biosynthesis in *Salvia*. To analyze the phylogenetic relationships of CYP76Ahs from different *Salvia* species. CYP76AH subfamily genes were screened from the previously obtained root transcriptomes of 48 *Salvia* plants, and SmCYP76AH1 was used as a template for homology screening. A total of 130 candidate CYP76AH genes with more than 55% homology and a longer than 1400 bp expression frame were obtained. A phylogenetic tree was constructed to preliminarily investigate the potential catalytic function of the candidate genes. Eleven functionally screened CYP76AH subfamily genes were downloaded, and CYP76AK1 was used as the root of the phylogenetic tree (Figure 3).

The discovered genes were named CYP76AH1, CYP76AH3, CYP76AH22-24, and CYP76AH30, with their species abbreviation prefixes based on gene sequence similarity. According to the clustering results, the CYP76AH1 and CYP76AH30 genes were clustered into one branch. The CYP76AH3 and CYP76AH22-24 genes clustered into a large branch, and the number of genes was significantly higher than that of other types of genes (Figure 3). The CYP76AH genes from *Salvia* plants were closely related to CYP76AH1, CYP76AH3, CYP76AH24, CYP76AH22, and CYP76AH23 but distantly related to CYP76AH11, CYP76AH8, CYP76AH11, CYP76AH15, CYP76AH16, and CYP76AH17.

The genetic distances between the species were calculated. The number of bootstrap replications was set to 50, while the gamma parameter was set to 1.00. The overall distance from *Salvia* was 0.20. The best maximum likelihood fits of the 24 nucleotide substitution models were selected. The BIC score was 35,910.78 for TVM + F + I + G4. Subsequently, the Xia test was performed using DAMBE 5.3.8, Iss < Iss.c, with Prob (two-tailed) being 0. CYP76AH was divided into five clades. Clade I included CYP76AH1 from *Salvia*. Clade II included CYP76AH22 and CYP76AH23 from *Salvia*. Clade III included CYP76AH24 from *Salvia*. Clade IV included CYP76AH24 from *Salvia*. Clade V was the root of the phylogenetic tree (Figure 3).

### 2.3. Biochemical Characterization of CYP76Ahs

Ten *Salvia* plants with a high abundance of tanshinones in the study of metabolic components were selected for CYP76AH gene cloning and functional characterization (Figure 2). Fifteen CYP76AH genes were cloned as candidate genes for functional research. To investigate the biochemical activity of these CYP76Ahs, recombinant expression in yeast (*Saccharomyces cerevisiae*) was employed [7,8]. Full-length cDNA of all CYP76Hs was cloned into the yeast expression vector pESC-His, and the resulting constructs were transformed into the WAT11 yeast strain in which the endogenous NADPH-CYP reductase was replaced by one from *Arabidopsis thaliana* [15]. In vitro assays were then carried out with microsomal preparations from induced cultures of this recombinant yeast, using miltiradiene as a substrate.

GC-MS and UPLC-Qtof-MS were employed to detect the enzyme reaction products. All 15 CYP76AH proteins from different *Salvia* plants had the ability to catalyze the formation of ferruginol from miltiradiene, but the conversion efficiency of the products varied (Figure 4A). Moreover, due to the heterogeneric catalytic function of CYP76AH subfamily proteins, ferruginol was further oxidized at other carbon sites [16]. Therefore, this study simultaneously detected other C7 and C11 oxidation products through UPLC-Qtof-MS (Figure 4B). Most CYP76AH proteins catalyze the continuous oxidation of ferruginol to produce sugiol, 11-hydroxyferruginol, and 11-hydroxy sugiol. However, there were significant differences in the amount of each product catalyzed by different CYP76AH subfamily proteins.

The main products generated by the catalytic reaction of SaeCYP76AH3, ScaCYP76AH3, SdaCYP76AH1, SdaCYP76AH3, ShoCYP76AH1, ShyCYP76AH3, SjaCYP76AH1, SjaCYP76AH3, SprCYP76AH1, and StrCYP76AH1 with miltiradiene were ferruginol and sugiol, while the production of 11-hydroxyferruginol and 11-hydroxy sugiol was relatively low or even undetectable. This indicates that the major catalytic sites of these CYP76AH subfamily proteins were C12 and C7 and that the hydroxylation ability for C11 was weak. Among them, the amounts of each product of SdaCYP76AH3 and SjaCYP76AH3 were relatively low. In addition, the main products of SflCYP76AH3, ShoCYP76AH3, SprCYP76AH3-01, SprCYP76AH3-02, and SseCYP76AH24 were 11-hydroxyferruginol and 11-hydroxysugiol, which directly convert a large amount of catalyzed ferruginol and sugiol, respectively, into their C11 hydroxylated products, indicating that these CYP76AH subfamily proteins may have higher C11 hydroxylation activity than others.

For further study on the catalytic efficiency of each CYP76AH, the same amount of each CYP76AH microsome and miltiradiene was added to the enzyme reaction, and SmCYP76AH1 and SmCYP76AH3 proteins from *Salvia miltiorrhiza* were used as controls. Relative quantitative analysis was performed using the amount of each product generated by SaeCYP76AH3 as one, and normalization analysis was performed based on the amount of each CYP76AH protein catalytic product (Figure 5A).

The results of the relative quantitative analysis showed that the main products of SflCYP76AH3, ShoCYP76AH3, SprCYP76AH3-01, SprCYP76AH3-02, and Sse CYP76AH24 were C11 hydroxylated products, with a product ratio consistent with SmCYP76AH3. Further analysis of these proteins and SmCYP76AH3 catalytic products revealed that the product yields of three proteins, SprCYP76AH3-01, SprCYP76AH3-02, and SseCYP76AH24, were significantly higher compared to SmCYP76AH3, especially the C11 hydroxylation products (Figure 5B). The production of 11-hydroxy ferruginol and 11-hydroxysugiol in the catalytic reaction of SprCYP76AH3-02 was 10.67 and 6.31 times higher than that of SmCYP76AH3, respectively.

### 2.4. Correlation Analysis of CYP76AH Protein Structure and Activity

To investigate the significant differences in the production of miltiradiene catalyzed by CYP76AH subfamily proteins, the amino acid sequences of each CYP76AH protein were compared. The sequence similarities of each protein with SmCYP76AH1, SmCYP76AH3, SpCYP76AH22, and SpCYP76AH24 were analyzed. Multiple CYP76AH sequences had a similarity of over 95% with the SmCYP76AH subfamily protein sequence (Table 1). Further analysis was conducted on the differential amino acid sites between these CYP76AH proteins and SmCYP76AHs. These CYP76AH genes have extremely high similarity and can be called natural multimutation genes of SmCYP76AH1 and SmCYP76AH3.

There was only one amino acid difference between ShyCYP76AH3 and SmCYP76AH1 in the N-terminal transmembrane domain, which had little effect on catalytic function; thus, there was no significant difference in its catalytic products (Table 1, Figure 5A). Amino acid sequence alignment analysis was conducted on each CYP76AH protein (Figure 6). Except for the N-terminal transmembrane domain, SjiaCYP76AH1 had only multiple consecutive amino acid sequences at 239–247 that differed from SmCYP76AH1 and were completely different from other CYP76AH proteins (Figure 6A,B). However, SjiaCYP76AH1 still catalyzed the production of four miltiradiene products, but its catalytic efficiency was significantly reduced compared to that of SmCYP76AH1.

Homologous modeling of the CYP76AH protein was performed using the Swiss model. Based on the SmCYP76AH1 protein crystal structure, the feasibility of homologous modeling of CYP76AH subfamily proteins was high. From the 3D structure of the protein, SjiaCYP76AH1 had multiple consecutive amino acid sequences at 239–247 that differed from SmCYP76AH1 and did not cause any changes in the active pocket structure of the protein. However, at the outer helical structure of the protein, changes in this continuous amino acid sequence caused twisting and folding in the middle of the helical structure (Figure 6A). From the analysis of amino acid properties, the change in the sequence at this location had little effect on the hydrophilicity of the protein, but it is possible that the protein substrate channel changed, affecting the ability of the protein to grasp the substrate miltiradiene, thereby reducing its catalytic efficiency compared to SmCYP76AH1.

Compared with SmCYP76AH1, StrCYP76AH1 lacked 43Q and had a Q277R difference. SprCYP76AH1 had an N279S difference, and ShoCYP76AH1 had an I489V difference (Figure 6C). It is possible that changes in amino acids in the external loose region affect the solubility of the protein, thereby affecting its catalytic efficiency and significantly reducing its catalytic product yield. Similarly, compared to all CYP76AH proteins in this study, SaeCYP76AH3 had a special 329R that differed from 329K, and ShoCYP76AH1 had a special 384M, while the others were 384K. These “natural mutations” occurred in the loose structural area of the outer layer of the protein, which may have affected its solubility and weakened its catalytic efficiency.

The C11 hydroxylation catalytic efficiency of SprCYP76AH3-01, SprCYP76AH3-02, and SseCYP76AH24 was significantly higher than that of SmCYP76AH3. The amino acid sequences of the substrate binding sites in their protein activity pockets were compared. These three proteins with significantly higher catalytic efficiency had certain differences in amino acids in their active pockets compared to the other CYP76AH subfamily proteins (Figure 6D). The amino acid sequences of 102E, 105G, 118G, and 307L differed from those of other CYP76AH proteins. In addition, multiple amino acid sites had CYP76AH1 and CYP76AH3 protein characteristics. For example, amino acids such as 97I, 116V, and 290T, were the same as those of the CYP76AH1 group, while amino acids such as 117G, 208S, 237Y, and 479F, had typical CYP76AH3 group protein sequence characteristics. The high catalytic efficiency of these CYP76AHs may be related to the advantageous amino acids that possess both CYP76AH1 and CYP76AH3 types of amino acids in the active pocket.

## 3. Discussion

In the *Salvia* genus, diterpenoid components are widely distributed and diverse; therefore, the proteins involved in the diterpenoid biosynthesis pathway are abundant and functionally diverse. In recent years, with the gradual discovery of functional genes in the biosynthetic pathway of plant-derived diterpenoids and the development of structural biology, an increasing number of protein structures have been analyzed and designed rationally or semirationally to obtain functionally modified mutants. However, the natural environment, the natural driving force behind mutation and evolution, may be a natural treasure trove for us to explore more efficient catalytic components.

In this study, three high-efficiency catalytic CYP76AH proteins were obtained, with significantly higher catalytic efficiency than the same functional proteins in widely cultivated *S. miltiorrhiza*. The modification of CYP76AHs in *S. miltiorrhiza* has been studied [16]. A series of mutant proteins with significantly higher catalytic efficiency was designed and obtained from wild-type SmCYP76AH1 and SmCYP76AH3 through homologous modeling and molecular docking analysis. Reasonably designed CYP76AH mutant protein microsomes catalyzed the production of multiple miltiradiene products, resulting in a significant increase in yield in engineered strains. Compared with the designed CYP76AH mutants, whose mutation sites were concentrated in the protein activity pocket, the amino acid closest to the substrate and heme molecules was in the substrate binding site (Figure 6D). Similar to the directed mutation design of the CYP76AH protein in *Salvia miltiorrhiza*, the selection of mutation sites for other CYP76AH proteins is also focused on the substrate binding region (Figure 6D), such as CYP76AH15 in *Coleus forskohlii* [17] and CYP76AH39 in *Salvia diorum* [18], both of which select mutation sites at key sites within 5 Å of the substrate and heme molecular spacing, such as 99A (100A in Salvia CYP76AHs), 236V (237V), 362G (363G), 366L, 367V (367L), and 479F (Figure 6D). However, these mutations often weaken or even completely abolish the function of the protein.

The amino acid differences between the CYP76AH mutant designed based on molecular docking and *Salvia* CYP76AH mining in this study are completely different. The sequence differences between SmCYP76AHs and Salvia CYP76AHs discovered in this article are concentrated in areas with relatively loose structures, such as protein outer walls or substrate channels. This may be due to the influence of differential amino acids on the solubility of proteins and the affinity of substrates entering the pocket. These differences can provide more reference for the rational design of such protein mutants.

## 4. Materials and Methods

### 4.1. Plant Materials and Chemicals

*Salvia* species were collected in September 2021. Both the leaves and roots of these species were obtained from the Shanghai Chenshan Botanical Garden (Shanghai, China). *Salvia* plant tissue was kept on dry ice. Then, the *Salvia* plants were snap frozen in liquid nitrogen and stored at −80 °C prior to total RNA extraction. Miltiradiene and ferruginol standards were purchased from Solarbio Bio-Technology (Beijing, China).

### 4.2. Chemical Constituents of Salvia

Standard amounts of 1.15 mg ferruginol, 1.20 mg sugiol, 1.13 mg miltirone, 1.18 mg cryptotanshinone, and 1.11 mg 15,16-dihydrotanshinone I were accurately weighed and placed in 2 mL centrifuge tubes. Two hundred microliters of chromatographic grade dimethyl sulfoxide were added for dissolution and mixed well after 45 min of ultrasound. The concentration of the prepared solution was 20 mmol/L. The dried *Salvia* (4 replicates for each species) sample roots were crushed with 80 mesh. A total of 0.02 g of plant root powder was accurately weighed, extracted with 1 mL of 70% methanol ultrasound for 45 min, and centrifuged (12,000 rpm min^−1^) for 15 min. The supernatant was pipetted, passed through a 0.22 µm filter membrane, immediately transferred to a liquid phase bottle, and stored at 4 °C. The chemical constituents of *Salvia* were analyzed by HPLC.

### 4.3. Identification of CYP76AH Genes

Ten *Salvia* species were chosen based on the metabolomic data. CYP76AH subfamily genes were screened from ten previously obtained *Salvia* plant transcriptomes. Using *SmCYP76AH1* as a template for homology screening, 420 genes with more than 55% homology were screened, including 130 CYP76AH genes with a gene expression frame length greater than 1400 bp measured in transcriptome data. Continuing to screen genes with a similarity greater than 75% to the SmCYP76AH1 sequence, a total of 15 candidate genes were obtained.

Total RNA was extracted from the bulb of *Salvia* with a Quick RNA Isolation Kit (Huayueyang Biotechnology, Beijing, China). The integrity of 1 μL of RNA was detected immediately by 1% agarose gel electrophoresis with 1 μL DNA/RNA loading buffer. RNA purity and concentration were gauged using a NanoDrop 2000 (Thermo Fisher Scientific, Waltham, MA, USA). Each species of *Salvia* cDNA was reverse transcribed using transcriptase (Jinsha Biotechnology, Beijing, China).

The full-length cDNA of *Salvia* was amplified using Phusion^®^ High-Fidelity DNA Polymerase (NEB, Ipswich, MA, USA) with the primers desighned. The reaction conditions were 45 s at 94 °C for denaturation, then 15 s at 94 °C, 15 s at 62 °C, 2 min at 72 °C for 38 cycles, and finally, 10 min at 72 °C extension. The cloned CYP76AH DNA was purified and collected at −20 °C.

### 4.4. Gene Diversity of CYP76AH Genes

A phylogenetic tree was constructed using the maximum likelihood (ML) method with 1000 bootstrap replications (Mega X). Base substitution saturation tests were performed (DAMBE5.3.8). The CYP76AH gene of *Salvia* showed high variability. There were 609 mutation sites and 78 haplotypes. The haplotype diversity was approximately 0.991, and the gene diversity was 0.10771 (DNASP 5.10).

### 4.5. Heterologous Expression of CYP76AH in Yeast and In Vitro Activity Assays

The recombinant plasmid pESC-His-CYP76AH was transformed into yeast strain WAT11 using a Frozen EZ Yeast Transformation II Kit™ (ZYMO, Irvine, CA, USA) for heterologous expression, and the WAT11 strain was transformed with the pESC-His empty vector control. Both were cultured in SD-His medium at 30 °C for 48 h. A single colony was selected and cultured for 48 h with an OD600 value greater than 2. Then, an equal volume of YPL (1% yeast extract, 2% peptone) was added for induction for 14 h, and the OD600 value reached 2–3. Combined with previous experimental experience, microsomes were chosen as the method for enzymatic reactions. The determination of in vitro activity was performed in a 500 μL reaction system, including 500 μM NADPH, 0.5 mg microsomal protein, 50 μM substrate, and a regenerating system, including 5 mM glucose-6-phosphate (with 1 unit glucose-6-phosphate dehydrogenase), 5 μM FAD, 5 μM FMN, and 1 μM DTT. The reaction was incubated at 30 °C and 200 rpm for 2.5 h. Then, 1000 μL ethyl acetate was used to extract the product (2 times). The first ethyl acetate extract (70 μL from 500 μL) was dispensed into gas-phase vials. The product was resolubilized with 70 μL of methanol.

### 4.6. Chemical Analysis of the Enzyme Reaction

Trace 1310 series GC and TSQ8000 MS were used to detect substrates and products (Thermo Fisher Scientific Co., Ltd., Waltham, MA, USA). Chromatographic separation was performed on a TR-5ms capillary column (30 m * 0.25 mm I.D. * 0.25 μm (film thickness dimension; Thermo Fisher Scientific, Waltham, MA, USA). The flow rate control was 1 mL min^−1^. 11-Hydroxyferruginol, sugiol, and 11-hydroxy sugiol were detected in vitro by UPLC-Qtof-MS (Waters Technologies, Milford, MA, USA). A BEH column (2.1 × 50 mm, 1.8 μm particle size; Waters Technologies, New York, NY, USA) was used for chromatographic separation. The mobile phase was acetonitrile and water (1% formic acid), with a UV absorption wavelength of 254 nm. Carnosic acid was used as an internal standard at 10 μM.

## 5. Conclusions

This study analyzed the functions of 15 CYP76AH genes derived from *Salvia* plants, all of which catalyzed the formation of ferruginol from miltiradiene. Some continued to catalyze the formation of three oxidation products at the C7 and C11 positions of ferruginol. By comparing the catalytic efficiency, three active catalytic elements with significantly improved catalytic efficiency for C11 hydroxylation were identified: SprCYP76AH3-01, SprCYP76AH3-02, and SseCYP76AH24. Through amino acid sequence alignment and molecular docking analysis of the CYP76AH protein in these *Salvia* plants, potential key amino acid residues, 117G, 208S, 237Y, and 479F, were identified. However, their impact on activity still requires further experimental verification.

*Salvia* plants containing tanshinones are a natural gene element library for mining functional genes in the tanshinone biosynthesis pathway. Compared to the CYP76AH subfamily genes with verified functions in *Salvia miltiorrhiza*, mining proteins with similar catalytic functions but higher expression levels in yeast and higher catalytic efficiency can provide efficient and advantageous natural mutant catalytic elements for the synthetic biology production of tanshinones.

## Figures and Tables

**Figure 1 molecules-28-04711-f001:**
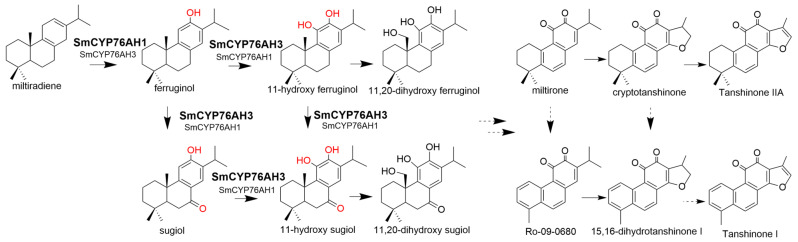
Biosynthetic pathway of tanshinones in *Salvia miltiorrhiza* mediated by CYP76AHs. The oxidation site of CYP76AHs in this study is marked in red, and the bold font represents the catalytic step of CYP76AHs.

**Figure 2 molecules-28-04711-f002:**
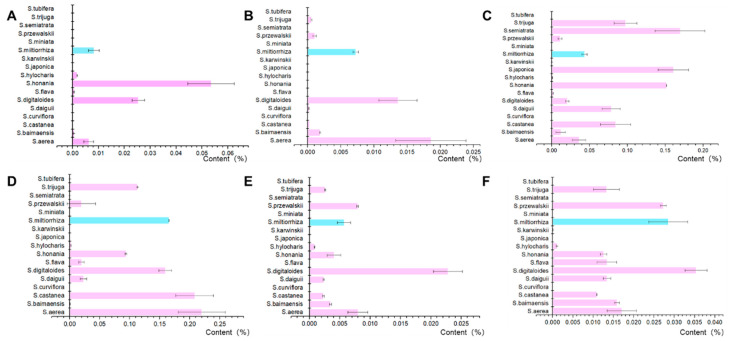
Content of key compounds in the tanshinone biosynthesis pathway in *Salvia* plants: ((**A**) ferruginol; (**B**) sugiol; (**C**) miltirone; (**D**) cryptotanshinone; (**E**) R09; and (**F**) dihydrotanshinone I). The blue column represents the content of compounds from *S. miltiorrhiza*, while the pink column represents the content of compounds in other *Salvia* species.

**Figure 3 molecules-28-04711-f003:**
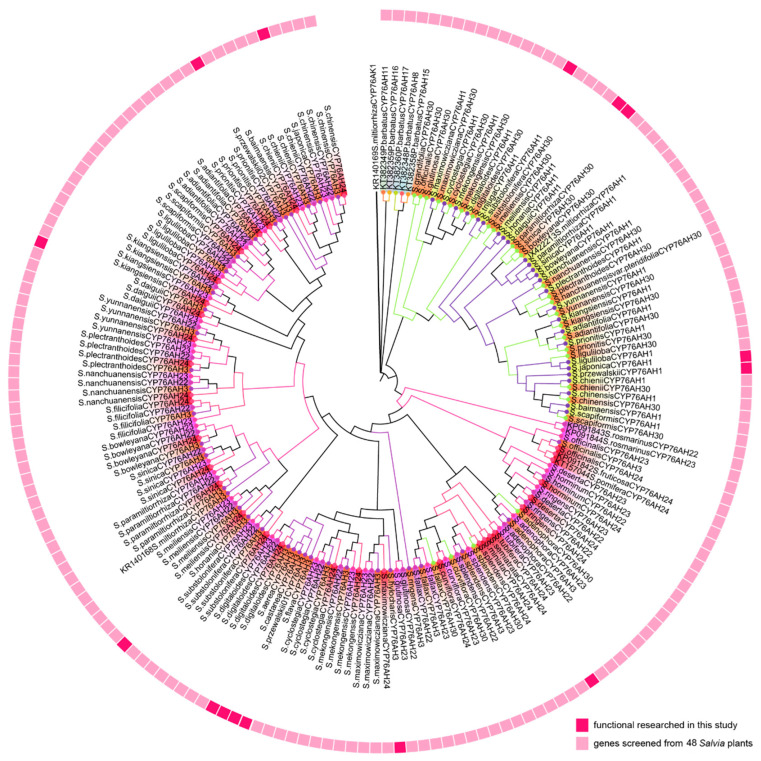
Phylogenetic tree of the *Salvia* CYP76AH subfamily genes. The same color indicates closer Homeotic gene. (GenBank accessions: SmCYP76H1, JX422213; SmCYP76AH3, KR140168; RoCYP76AH4; RoCYP76AH22, KP091843; RoCYP76AH23, KP091844; SfCYP76AH24, KP091842; SpCYP76AH24, KT157044S; PbCYP76AH8, KT382348; PbCYP76AH11, KT382349; PbCYP76AH15, KT382358; PbCYP76AH16, KT382359 and PbCYP76AH17, KT382360).

**Figure 4 molecules-28-04711-f004:**
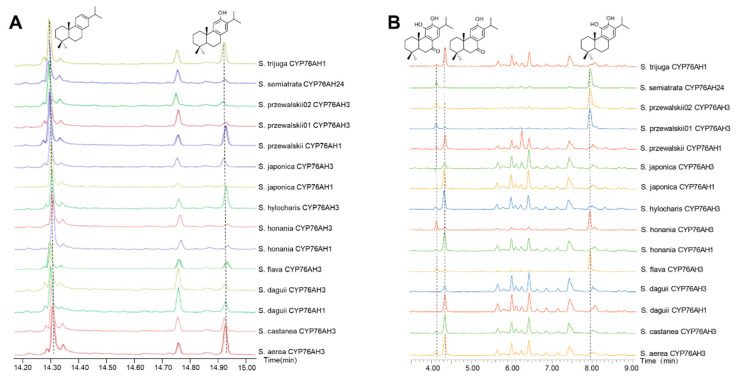
Chromatograms of reaction products of the CYP76AH protein with miltiradiene. The dashed line represents the peak position. (**A**) GC-MS chromatogram of miltiradiene and ferruginol; (**B**) UPLC-Qtof-MS chromatogram of 11-hydroxyferruginol, sugiol, and 11-hydroxysugiol.

**Figure 5 molecules-28-04711-f005:**
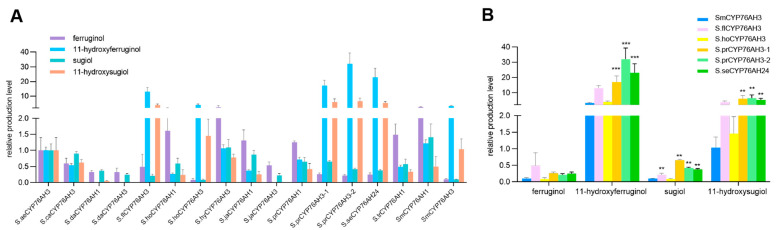
Relative quantitative analysis of the products from the reaction of miltiradiene catalyzed by CYP76AH microsomes. (**A**) Relative abundance of 15 Salvia CYP76AHs and SmCYP76AHs protein catalytic products; (**B**) comparison of efficiency between five CYP76AH proteins with high catalytic product levels and SmCYP76AH3. Error bars represent the standard deviation (SD) (*n* = 3 biologically independent samples; *** *p* < 0.001 by a two-sided Student’s *t*-test, while ** *p* < 0.01 by a two-sided Student’s *t*-test).

**Figure 6 molecules-28-04711-f006:**
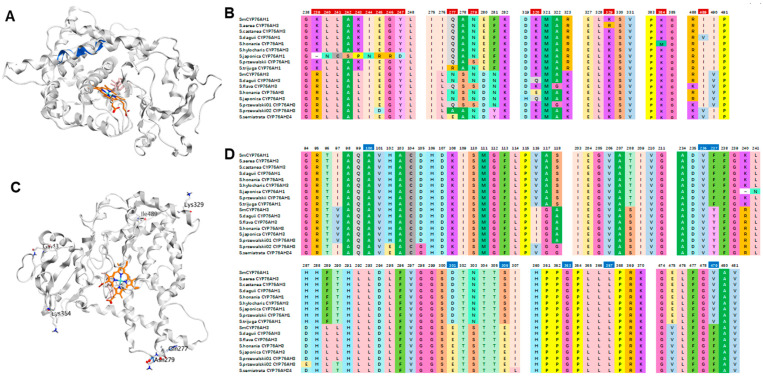
Amino acid and protein structure differences between SmCYP76AHs and CYP76AHs in *Salvia* species. (**A**) Protein model docking analysis of SjiaCYP76AH1 and SmCYP76AH1. (**B**) Partial differential amino acid sequences between SmCYP76AHs and *Salvia* CYP76AHs in this study. (**C**) Schematic diagram of amino acid differences between SmCYP76AH1 and Salvia CYP76AHs. (**D**) Comparison of amino acids at substrate binding sites of CYP76AH subfamily proteins. The SmCYP76AH protein structure model is shown in white, the heme is shown in orange, and the substrate is shown in pink. The differential amino acids of each Salvia CYP76AH are displayed in red, and the reported artificially designed directed evolution sites are displayed in blue.

**Table 1 molecules-28-04711-t001:** Sequence similarity of *Salvia* CYP76AHs and identified CYP76AHs.

	Amino Acid Similarity (%)				Sequence Differences	
	SmCYP76AH1	SmCYP76AH3	SpCYP76AH22	SpCYP76AH24	SmCYP76AH1	SmCYP76AH3
SaeCYP76AH3	99.60	79.60	78.26	77.73	S3Y, K329R	
ScaCYP76AH3	95.91	77.00	75.76	75.24	S3Y, KS494NPRIRNTTHYRARASTWNRS	
SdaCYP76AH1	98.99	78.99	77.67	77.14		
SdaCYP76AH3	80.00	99.39	86.93	86.85		S5P, Q161E, K320Q
SflCYP76AH3	79.39	96.76	87.33	86.85		S5P, F12L, S14T, S21F, S22F, R157K, Q161K, A258T, D261N, N277Q, N279S, V315I, A322G, A394G, D466N, R488K
ShoCYP76AH1	99.39	79.60	78.26	77.53	F4S, T18I, K384M	
ShoCYP76AH3	79.80	99.39	86.73	86.65		S3N, S5P, K320E
ShyCYP76AH3	99.80	79.80	78.46	77.93	S3I	
SjaCYP76AH1	98.99	79.60	77.43	76.49	D2E, F4S, I13T, T18I, K156R	
SjaCYP76AH3	80.00	98.58	86.93	86.65		S5P, H156D, R157K, D319H, K320Q, R345S, Q473G
SprCYP76AH1	99.19	79.39	78.06	77.53	D2E, F4S, N279S	
SprCYP76AH3-01	80.30	96.76	87.52	87.65		S3Y, S5P, S14T, S21F, S22F, R157K, Q161K, D261N, N277Q, N279S, A322G, D406E, A407S, Q409E, D466N, R488K
SprCYP76AH3-02	76.67	85.49	83.88	83.23		
SseCYP76AH24	79.39	79.60	86.93	86.25		

## Data Availability

Not applicable.

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
