# Peer review of "Functional Study and Efficient Catalytic Element Mining of CYP76AHs in Salvia Plants"

_molecules, 2023, doi:10.3390/molecules28124711_

Round 1

Reviewer 1 Report

The CYP76AH subfamily (P450) played important roles in synthesis of tanshinones, which are exclusive compounds in the Salvia. The study of CYP76AH genes connected the sequence evolution and catalytic efficiency, which identified structure-function relationship and revealed some amino acids charging for unique enzyme function, in excellent quality.

Some minor suggestions:

Line 80: more detailed figure legend to describe the reaction process.

Line 106: Figure legend needs more detailed information.

Line 129: Figure 3. Figure 3. Double Figure 3 and delete one copy.

Line 165: Fig 4 need to change to Figure 4.

Line 199: Table 1 would be better placed in the one page.

Line 223: Figure legend needs more detailed to description.

After corrections as suggested, I recommend this manuscript accepted.

Author Response

Dear Editor and Reviewers,

We appreciate all of your constructive points and suggestions and have modified our manuscript accordingly. The language of the article has been polished by the company, and the modified parts have not been marked in red due to excessive polishing. Below, we detail the changes made in response to the specific points made by the reviewers, and the modifications to the research content are marked in red.
Reviewer 1:

Point 1: Line 80: more detailed figure legend to describe the reaction process.

Response 1: We appreciate your suggestion. We have revised Figure 1 and show more information in the figure legend.

Point 2: Line 106: Figure legend needs more detailed information.

Response 2: Thank you for your suggestions. More information is shown in the figure legend and colored in red.

Point 3: Line 129: Figure 3. Figure 3. Double Figure 3 and delete one copy.

Response 3: Thank you very much for your Points. The mistake was modified.

Point 4: Line 165: Fig 4 need to change to Figure 4.

Response 4: Thank you very much for your Points. The mistake was modified.

Point 5: Line 199: Table 1 would be better placed in the one page.

Response 5: Thank you for your suggestions. Table 1 was modified on one page.

Point 6: Line 223: Figure legend needs more detailed to description.

Response 6: We have revised Figure 6 and shown more information in the figure legend.

We hope you find the revised manuscript acceptable.

With best regards,

Ying Ma

Reviewer 2 Report

1)      Line No. 89: 17 distinct species and not 16 as per Fig 2

2)      In fig 2A) S. digitaloiides represents instead of S. daiguii (0.02534%)

3)      Table 1. Sequence spelling mistake in heading

4)      Refer 12 and 13 rows of Table1. SprCYP76AH3- 02 is shown twice in 12 &13 rows instead of SprCYP76AH3- 01

5)      In line No.183, SmCYP76AH1 is also given as control. In fig 5. SmCYP76AH1 control and CYP76AH1 genes of different salvia species are also not shown.

6)      In table 1. Add Sm in first and second column heading

7)      Refer line No.215, there is no ShyCYP76AH3 in Fig5

8)      In conclusion line No. 363, this study analysed the functions of 11 CYP76AH genes, but in Fig3, it is shown that functional researched in this study are 15 and in chromatogram protein study also 15 is there, then why only 11 genes functional analysis in conclusion.

Only minor editing of English language is required.

Author Response

Dear Editor and Reviewers,

We appreciate all of your constructive points and suggestions and have modified our manuscript accordingly. The language of the article has been polished by the company, and the modified parts have not been marked in red due to excessive polishing. Below, we detail the changes made in response to the specific points made by the reviewers, and the modifications to the research content are marked in red.

Point 1: Line No. 89: 17 distinct species and not 16 as per Fig 2
Response 1: Thank you for your Points. Sixteen Salvia species were studied, while S. miltiorrhiza was used as a control.

Point 2: In fig 2A S. digitaloiides represents instead of S. daiguii (0.02534%)
Response 2: Thank you very much for your Points. The mistake was modified in the manuscript.

Point 3: Table 1. Sequence spelling mistake in heading
Response 3: Thank you for your suggestion. The spelling mistake was modified.

Point 4: Refer 12 and 13 rows of Table1. SprCYP76AH3- 02 is shown twice in 12 &13 rows instead of SprCYP76AH3- 01

Response 4: Thank you very much for your Points. This mistake was modified in the manuscript.

Point 5: In line No.183, SmCYP76AH1 is also given as control. In fig 5. SmCYP76AH1 control and CYP76AH1 genes of different salvia species are also not shown.

Response 5: Thank you for your suggestion. We encountered a huge mistake when uploading the manuscript. The order of Figures 5 and 6 was incorrect. We integrated the two images into Figure 5A, B and cited them in the text.

Point 6: In table 1. Add Sm in first and second column heading
Response 6: Thank you very much for your Point. The plant name was added in the appropriate column of Table 1.

Point 7: Refer line No.215, there is no ShyCYP76AH3 in Fig5

Response 7: Thank you very much for your Point. The figure was revised.

Point 8: In conclusion line No. 363, this study analysed the functions of 11 CYP76AH genes, but in Fig3, it is shown that functional researched in this study are 15 and in chromatogram protein study also 15 is there, then why only 11 genes functional analysis in conclusion

Response 8: Thank you very much for your Point. This mistake was corrected.

We hope you find the revised manuscript acceptable.

With best regards,

Ying Ma

Reviewer 3 Report

 This manuscript studies 15 CYP76AH genes from 10 Salvia species that were cloned and studied from evolution and catalytic efficiency perspectives. Structure-function relationship revealed several conserved residues which might be related to the function of CYP76AHs.

There are important flaws in the manuscript listed below:

1-    The manuscript has major grammar and punctuation problems. It needs to be checked for the English language by a native speaker.

2-         The title and abstract could be changed to better match the text;

3-       Materials and methods section should be expanded and more details included, mainly in the Identification of CYP76AH genes (4.3);

4-    This discussion doesn't align with the study - it should be revised based on the outcomes of the experimental work and compare with previous research;

5-    It is better to replace some unrelated references in the manuscript with some new publications.

The manuscript has major grammar and punctuation problems. It needs to be checked for the English language by a native speaker.

Author Response

Dear Editor and Reviewers,

We appreciate all of your constructive points and suggestions and have modified our manuscript accordingly. The language of the article has been polished by the company, and the modified parts have not been marked in red due to excessive polishing. Below, we detail the changes made in response to the specific points made by the reviewers, and the modifications to the research content are marked in red.

Point 1: The manuscript has major grammar and punctuation problems. It needs to be checked for the English language by a native speaker.
Response 1: We appreciate your suggestion. Our article's language and grammar have been carefully checked. Moreover, the language of the article was edited by the company.

Point 2: The title and abstract could be changed to better match the text;

Response 2: The title of the article has been changed to “Functional Study and Efficient Catalytic Element Mining of CYP76AHs in Salvia Plants”. The abstract of the article has been revised.

Point 3: Materials and methods section should be expanded and more details included, mainly in the Identification of CYP76AH genes (4.3);
Response 3: Thank you very much for your suggestion. We have added a detailed introduction to the materials and methods.

Point 4:

This discussion doesn't align with the study - it should be revised based on the outcomes of the experimental work and compare with previous research;
Response 4: We appreciate your suggestion. The introduction to the Salvia genus has been removed, and the differences between the components excavated in this article and the manually designed CYP76AH components have been discussed.

Point 5: It is better to replace some unrelated references in the manuscript with some new publications.
Response 5: Thank you very much for your suggestion. We have revised the references and highlighted them in red.

We hope you find the revised manuscript acceptable.

With best regards,

Ying Ma

Round 2

Reviewer 3 Report

The authors revised the manuscript according to the referee's comments; therefore, the manuscript could be accepted.